# Quantifying alkane emissions in the Eagle Ford Shale using boundary layer enhancement

Geoffrey Roest[1] and Gunnar Schade[1]

[1]Department of Atmospheric Sciences, Texas A&M University, 3150 TAMU, College Station, Texas 77843-3150

*Correspondence to:* Geoffrey Roest (gsroest@tamu.edu)

**Abstract.** The Eagle Ford Shale in southern Texas is home to a booming unconventional oil and gas industry, the climate and air quality impacts of which remain poorly quantified due to uncertain emissions estimates. We used the atmospheric enhancement of alkanes from Texas Commission on Environmental Quality volatile organic compound monitors across the shale, in combination with back trajectory and dispersion modeling, to quantify $C_2$-$C_4$ alkane emissions for a region in southern Texas, including the core of the Eagle Ford, for a set of 68 days from July 2013 to December 2015. Emissions were partitioned into raw natural gas and liquid storage tank sources using gas and headspace composition data, respectively, and observed enhancement ratios. We also estimate methane emissions based on typical ethane-to-methane ratios in gaseous emissions. The median emission rate from raw natural gas sources in the shale, calculated as a percentage of the total produced natural gas in the upwind region, was 0.7% with an interquartile range (IQR) of 0.5% -1.3%, below the U.S. Environmental Protection Agency's (EPA) current estimates. However, storage tanks contributed 17% of methane emissions, 55% of ethane, 82% percent of propane, 90% of *n*-butane, and 83% of isobutane emissions. The inclusion of liquid storage tank emissions results in a median emission rate of 1.0% (IQR of 0.7-1.6%) relative to produced natural gas, overlapping the current EPA estimate of roughly 1.6%. We conclude that emissions from liquid storage tanks are likely a major source for the observed non-methane hydrocarbon enhancements in the northern hemisphere.

## 1 Introduction

The recent boom in onshore oil and gas production in the U.S. has heightened concerns over the environmental impacts of petroleum as an energy source. Technological advances in petroleum recovery methods, namely hydraulic fracturing and horizontal drilling (hereafter referred to as unconventional oil and gas or UOG development), have allowed for the production of previously inaccessible hydrocarbons in shale plays and tight sand formations. Emissions from the rapidly changing infrastructure in U.S. oil and gas fields have introduced new uncertainties in the climatological and air quality impacts associated with petroleum production. However, independent scientific studies of these impacts are sparse in some shale plays that have developed rapidly, including the Eagle Ford Shale (EFS) in southern Texas.

Emissions from oil and gas production include methane, a potent greenhouse gas (GHG); non-methane volatile organic compounds (NMVOCs) and oxides of nitrogen ($NO_x$), important precursors for ozone formation; carbon monoxide; and hazardous air pollutants (HAPs) such as benzene, a known carcinogen (*U.S. Centers for Disease Control*, 2014). There are a variety of

emission sources for each of these trace gases during oil and gas exploration, production, and distribution, resulting in co-emissions of GHGs, NMVOCs, $NO_x$, and HAPs. Direct emissions from oil and gas operations include hydrocarbons from flowback events and well completions, on-site equipment during the routine operation of the well, such as compressors and storage tanks, and unintentional emissions such as leaks from valves and pipelines. Indirect emissions of NMVOCs, HAPs,

and $NO_x$ come largely from combustion sources such as diesel engines in generators, drilling rigs, compressors, and trucks used to transport equipment, water, and petroleum (*Field et al.*, 2014). In addition, flaring is widely used in certain shale plays, including the EFS, to dispose of excess natural gas – mostly *associated gas* produced at oil wells (*Tedesco and Hiller*, 2014). Flaring is intended to efficiently dispose of hydrocarbons, resulting in emissions of carbon dioxide as opposed to methane and higher hydrocarbons, including HAPs. However, the high temperatures in flares result in $NO_x$ emissions, and inefficiencient

flaring has been shown to release unburned hydrocarbons and NMVOCs produced via pyrolysis (*Strosher*, 2000; *Olaguer*, 2012; *Pikelnaya et al.*, 2013).

The regional and global impacts of methane emissions from UOG development in the U.S. are poorly quantified due to widely varying and largely uncertain emissions estimates. According to the U.S. Environmental Protection Agency's (EPA) 2016 greenhouse gas inventory (*U.S. Environmental Protection Agency*, 2016), approximately 6,570 Gg of methane was emitted

from all oil and gas field production in the U.S. and in offshore federal waters in 2011 when voluntary emissions reductions are included. This corresponds to $12 \times 10^9$ m$^3$ of natural gas at 1 atm and 15°C, assuming an average methane content of 80% by volume in raw U.S. natural gas (*Pétron et al.*, 2012). During the same year, nationwide gross natural gas production from oil and gas wells totaled $756 \times 10^9$ m$^3$ (*U.S. Energy Information Administration*, 2017). Therefore, the EPA's emission rate of natural gas from oil and gas field production in the U.S. in 2011 was 1.6% relative to the volume of produced natural gas. The

EPA's 2016 greenhouse gas inventory estimated methane emissions of 6,985 Gg in 2013, which, when compared to the 2013 U.S. natual gas production of $853 \times 10^9$ m$^3$, yields a relative emission rate of 1.5%, a slight reduction from 2011. However, the EPA's methane emission estimates from U.S. oil and gas field production in the 2016 greenhouse gas inventory increased from the 2015 EPA greenhouse gas inventory (*U.S. Environmental Protection Agency*, 2015a). In the 2015 greenhouse gas inventory, the estimated methane emissions from U.S. oil and gas field production for 2013 totaled 2,722 Gg or 0.6% of produced natural

gas by volume. The increase between the 2015 and 2016 greenhouse gas inventories is due, in part, to updated emission factors for methane emissions.

Several recent top-down estimates of emissions from individual shale plays exceeded bottom-up inventories, at times by an order of magnitude or more (*Karion et al.*, 2013; *Pétron et al.*, 2012, 2014; *Miller et al.*, 2013). Several aircraft measurements performed upwind and downwind of major natural gas-producing shale areas (*Peischl et al.*, 2015) suggest that

higher-than-inventory emissions in one shale area might be compensated by lower-than-inventory emissions in another shale area, suggesting that national emissions might be represented correctly in the inventory. More recently, a series of studies in the Barnett Shale, which mostly produces natural gas, have attempted to reconcile the differences between bottom-up and top-down estimates (e.g. *Karion et al.*, 2015; *Lyon et al.*, 2015; *Smith et al.*, 2015). While the emissions estimates from the Barnett Shale studies were lower than other top down estimates, the studies concluded (*Zavala-Araiza et al.*, 2015) that cur-

rent emissions inventories, such as the EPA's Greenhouse Gas Reporting Program (GHGRP) (*U.S. Environmental Protection*

*Agency*) and the Emission Database for Global Atmospheric Research (EDGAR) (*European Commission, Joint Research Centre (JRC)/Netherlands Environmental Assessment Agency (PBL)*, 2009), underestimate emissions from oil and gas production in the Barnett Shale, similar to a prior conclusion reached by *Zavala-Araiza et al.* (2014) based on hydrocarbon measurements collected throughout the shale area by the Texas Commission on Environmental Quality (TCEQ). Furthermore, a recent study
using satellite measurements of tropospheric methane (*Turner et al.*, 2016) concluded that North American methane emissions have increased by approximately 30% between 2002 and 2014, most likely as a result of UOG development. In combination, these studies strongly suggest that bottom-up inventories underestimate actual emissions from oil and gas systems.

   The uncertainty and variability in methane emission estimates suggest that NMVOC emissions are also poorly quantified, as methane and NMVOCs are often co-emitted. The EFS may have particularly poorly quantified emission estimates of both
methane and NMVOCs due to the widespread use of flaring (*Tedesco and Hiller*, 2014). The region has experienced a boom in UOG production in the region since 2008 (*Gebrekidan*, 2011)], and the EFS is currently one of the most productive shale plays in the United States (*U.S. Energy Information Administration*, 2016). In nearby San Antonio, Bexar County, ozone mixing ratios increased in tandem with oil and gas production rates during the initial growth years of the EFS (*Schade and Roest*, 2015). The increasing ozone levels have raised concerns over public health in the city, which is in danger of being designated
as a nonattainment area by the EPA. Bexar County ozone design values for recent years exceeded both the current and previous ozone standards of 70 and 75 ppb (*Alamo Area Council of Governments (AACOG)*, 2014; *U.S. Environmental Protection Agency*, 2015b), while ozone levels in Houston, a non-attainment area, have dropped below levels in San Antonio. Modelling studies for the San Antonio region (*Alamo Area Council of Governments (AACOG)*, 2013a; *Pacsi et al.*, 2015) have found that emissions from the EFS contribute significantly to regional ozone formation. However, these studies have utilized emissions
inventories that may underestimate the magnitude of emissions from oil and gas operations in the shale area. *Schneising et al.* (2014) estimated methane emissions in the western EFS using trends in satellite based measurements of total column methane from 2006 to 2008 compared with 2009 to 2011. Their study yielded an emission rate of 9.1% (2.9% – 15.3%), expressed as a fraction of increased methane emissions vs. increases in produced energy in the region. While their study focused on methane emissions, the co-emission of methane and NMVOCs suggests that existing VOC inventories may also be biased low,
and the emissions of VOCs would result in the additional loss of produced energy. In another satellite data study, *Duncan et al.* (2016) showed that tropospheric $NO_2$ concentrations have increased in shale areas that practice flaring, including the EFS, while concentrations have decreased in US urban areas. Since most of the EFS is part of rural south Texas where $NO_x$ concentrations are comparatively low, increases in $NO_x$ can contribute strongly to ozone formation. Therefore, the increase in ozone associated with ozone precursor emissions from the EFS may be higher than what the existing ozone modeling studies
suggest.

   This study uses the atmospheric enhancement of short-chain alkanes – ethane ($C_2$), propane ($C_3$), isobutane ($iC_4$), *n*-butane ($nC_4$), isopentane ($iC_5$), and *n*-pentane ($nC_5$) – between upwind and downwind measurement locations to estimate alkane emissions from a region in southeast Texas including the core of the EFS. Alkanes dominate atmospheric OH radical reactivity at a TCEQ monitoring site north of the EFS (*Schade and Roest*, 2016) and the emission estimates for these short-chain alkanes
are needed to assess the potential air quality impacts from the EFS. We focus on ethane as a tracer for oil and gas emissions as

it is the second most abundant compound in natural gas (*Xiao et al.*, 2008) and, unlike methane, it is not emitted by microbial sources in significant quantities (*Simpson et al.*, 2012). Recent increases in ethane abundance in the northern hemisphere have been linked to UOG production in the U.S. (*Franco et al.*, 2016; *Helmig et al.*, 2016; *Kort et al.*, 2016). Ethane has thus been used in previous oil and gas emissions estimates (*Schwietzke et al.*, 2014; *Smith et al.*, 2015), and statistically significant increases in ethane mixing ratios have been observed downwind of the EFS during its development (*Schade and Roest*, 2015). The $C_3$ and $C_4$ alkane-to-ethane enhancement ratios are used to estimate the relative contributions of raw natural gas emissions and vented gases from liquid storage tanks, two major sources of gaseous emissions from upstream UOG that have varying compositions (*Brantley et al.*, 2014; *Field et al.*, 2014; *Lyon et al.*, 2015; *Kort et al.*, 2016). A mass balance approach and a Monte Carlo simulation are then used to estimate the emissions of $C_2$-$C_4$ alkane from raw natural gas emissions and liquid storage tank venting and the associated uncertainties. Methane emissions are also estimated using methane-to-ethane ratios in raw natural gas and vented storage tank gas. Lastly, the methane emission rate is expressed as a fraction of the produced natural gas to compare our emission estimate with other top-down studies.

## 2 Methods

### 2.1 TCEQ Data

The TCEQ operates a network of air quality monitoring sites across the state of Texas, some of which measure NMVOCs including alkanes, alkenes, cycloalkanes, and aromatics. The TCEQ collects NMVOC data to the east and southeast of the EFS in Clute and at several sites in Corpus Christi, including "Hillcrest" and "Oak Park" (Fig. 1), which were selected for use in this study due to data availability and their location. Other sites in Corpus Christi are immediately downwind of major local point sources when winds are blowing from the Gulf of Mexico. To the northwest of the EFS, NMVOC data have been collected since summer 2013 in Floresville, a small city immediately north of the shale area, and in northwest San Antonio ("Old Highway 90"). Descriptions of the five sites that are used in this study are presented in Table 1. Data from these sites have been previously used to demonstrate trends in ethane mixing ratios near the EFS (*Schade and Roest*, 2015). The emissions estimates in this study were performed using hourly ethane data from the automated ozone precursor monitoring sites in Floresville and Corpus Christi – Oak Park (hereafter referred to as "Oak Park"). While the Oak Park site was installed before the oil and gas boom in the EFS, data for Floresville are only available since 19 July 2013. Therefore, direct data comparisons were performed only for a thirty-month period from July 2013 through December 2015.

Alkane mixing ratios at Floresville and Oak Park were compared under south to southeasterly flow regimes, when Corpus Christi is upwind of the EFS and Floresville is downwind. South to southeasterly flow regimes were identified using 48 hour back-trajectories originating at Floresville from the Hybrid Single-Particle Lagrangian Integrated Trajectory (HYSPLIT) model (*Stein et al.*, 2015). The trajectories were run four times per day at an interval of six hours beginning at 06:00 UTC (00:00 LST). The EDAS 40 km dataset (*National Centers for Environmental Prediction (NCEP)*) was used for meteorology in the HYSPLIT model. This dataset, which adequately captures synoptic scale flow, was chosen for computational efficiency. A series of polygons, as discussed in supporting Text S1 and Table S1 and shown in supporting Fig. S1, were used to identify air

mass origins. Trajectories that had continental origins during the previous 48 hours were flagged in order to isolate those with marine origins. Only days with at least three out of four southeasterly trajectories and at least 75% completeness (i.e., at least eighteen hours of NMVOC data) at both Floresville and Oak Park were used.

The TCEQ sites in Floresville and Oak Park measure hydrocarbons using nearly identical automated GC-FID systems which record continuous hourly data from 40 min, 600 mL air samples. Standard operating procedures for these instruments are available from the Field Operations Division of the TCEQ (*Texas Commission on Environmental Quality*, 2005). The method detection limit is 0.4 ppbC, and instrument precision is measured using weekly injections of propane and benzene standard gases. Data are quality assured by the TCEQ if the relative difference between standard gas measurements remains less than 20%. We have assumed that this value is representative of the two-standard-deviation uncertainty of an individual measurement. The mean afternoon alkane mixing ratios for each day were calculated at both sites by averaging hourly mixing ratios during the afternoon hours (15:00 to 18:00 LST), when daytime convection allows for mixing throughout the planetary boundary layer (*Stull*, 2009). The alkane enhancement for each day was determined by subtracting the mean of three afternoon alkane mixing ratios at the upwind site of Oak Park from the mean afternoon mixing ratio at the downwind site in Floresville. The relative standard error of the three hourly measurements at each site is 5.8%, and the uncertainty in the enhancement is equal to the sum of the absolute errors of the afternoon mixing ratios at each site.

## 2.2 Alkane Sources

In this study, we assumed that regional alkane emissions are dominated by UOG operations in the EFS. Other sources of ethane emissions were assumed to be negligible as no regional biomass burning was reported during the study period (*Randerson et al.*, 2015). However, the mixing ratios of longer chain alkanes (notably $C_5$ and higher) may be influenced by evaporative and tailpipe emissions from nearby automotive traffic (*Tsai et al.*, 2006; *Ho et al.*, 2009; *Simpson et al.*, 2012). Emissions from UOG come from multiple sources and can include emissions of raw natural gas from compressors, flowback events, and unintentional leaks. The composition of these gases are dominated by the most volatile hydrocarbons, i.e. methane and ethane. In comparison, emissions from storage tanks, used to store liquids from wells prior to transportation and further processing, have been shown to contribute largely to hydrocarbon emissions in UOG shale plays (*Lyon et al.*, 2015, 2016). Since gas produced at the well is separated from liquids prior to storage, the headspace in storage tanks is primarily composed of hydrocarbons heavier than ethane, notably short-chain alkanes such as propane, butanes, and pentanes, although methane and ethane may still be present. We assume that regional short-chain alkane emissions are dominated by gases produced at the wellhead (referred to as *raw natural gas*) and emissions from liquid storage tanks (referred to as *tank gas*). Table 2 shows the available compositions of raw natural gas samples from the EFS and tank gas samples from the Barnett Shale. To our knowledge, no tank gas composition data are publicly available for the EFS. The sampled emissions from liquid storage tanks in the Barnett Shale are variable in composition and this is incorporated into our error analysis. The composition of emissions from oil and condensate storage tanks in conventional production areas in Texas (*Hendler et al.*, 2009) are also largely variable. We assume that the average composition of liquid storage tank emissions in the Eagle Ford falls within the variability of the Barnett Shale samples, although this assumption introduces an unquantified source of uncertainty in our analysis.

Observed alkane enhancement ratios can be partitioned into emissions from multiple sources, including raw natural gas and tank gas emissions. Equation 1 shows the partitioning of (e.g.) observed propane-to-ethane ratios into raw natural gas, tank gas, and all other sources.

$$\left(\frac{C_3}{C_2}\right)_{observed} = f_{RNG}\left(\frac{C_3}{C_2}\right)_{RNG} + f_{TG}\left(\frac{C_3}{C_2}\right)_{TG} + f_{other}\left(\frac{C_3}{C_2}\right)_{other} \tag{1}$$

where $\left(\frac{C_3}{C_2}\right)_{RNG}$, $\left(\frac{C_3}{C_2}\right)_{TG}$, and $\left(\frac{C_3}{C_2}\right)_{other}$ represent the $C_3/C_2$ ratios in emissions from raw natural gas, tank gas, and other sources, respectivey, and the relative contributions to the observed ratio from each source ($f_{RNG}$, $f_{TG}$, and $f_{other}$) add up to 1. If raw natural gas and tank gas sources dominate regional alkane emissions and other sources can be considered negligible, then $f_{RNG} + f_{TG} \approx 1$ and

$$\left(\frac{C_3}{C_2}\right)_{observed} = f_{RNG}\left(\frac{C_3}{C_2}\right)_{RNG} + (1 - f_{RNG})\left(\frac{C_3}{C_2}\right)_{TG} \tag{2}$$

$$f_{RNG} = \frac{\left(\frac{C_3}{C_2}\right)_{observed} - \left(\frac{C_3}{C_2}\right)_{TG}}{\left(\frac{C_3}{C_2}\right)_{RNG} - \left(\frac{C_3}{C_2}\right)_{TG}} \tag{3}$$

Here, $f_{RNG}$ is found using $C_3/C_2$ and verified using $iC_4/C_2$ or $nC_4/C_2$. ratios. This number represents the fraction of ethane attributed to emissions from raw natural gas sources, such that $C_{2,RNG} = f_{RNG} \cdot C_{2,observed}$ and $C_{2,TG} = (1 - f_{RNG}) \cdot C_{2,observed}$. The expected methane enhancement can be estimated using eq. 4.

$$C_1 = C_{2,observed}\left( f_{RNG}\left(\frac{C_1}{C_2}\right)_{RNG} + (1 - f_{RNG})\left(\frac{C_1}{C_2}\right)_{TG} \right) \tag{4}$$

Similarly, the methane enhancement estimate, along with other alkanes, can be attributed to raw natural gas and tank gas sources as follows.

$$C_{1,RNG} = C_{2,RNG} \cdot \left(\frac{C_1}{C_2}\right)_{RNG} \tag{5}$$

$$C_{1,TG} = C_{2,TG} \cdot \left(\frac{C_1}{C_2}\right)_{TG} \tag{6}$$

## 2.3   Mass Balance Approach

Short-chain alkane emissions from a region encompassing the central section of the EFS were quantified using a mass-balance approach that has been adapted to an area source (eq. 7), in which emissions are considered to be spatially and temporally homogenous.

$$F = \left(\bar{U} \cdot \cos\alpha\right) \cdot \bar{n} \cdot \int_{Z_0}^{Z_{PBL}} \rho(z)\,dz \cdot \Delta x \tag{7}$$

A different form of this mass-balance method has been used in previous emissions estimates for emission plumes from oil and gas systems (*Karion et al.*, 2013, 2015; *Caulton et al.*, 2014; *Pétron et al.*, 2014; *Smith et al.*, 2015). This estimate can be biased low as it does not account for the entrainment of air from the free troposphere into the planetary boundary layer (PBL) (*Karion et al.*, 2015), but can also be biased high if nearby emissions produced unmixed plumes. We consider the Floresville site to be sufficiently downwind of ethane sources such that it is not impacted by discrete plumes if the PBL is well mixed. In our approach, we assume that the component of the wind that is parallel to the transect between upwind and downwind measurement sites ($\bar{U} \cdot \cos \alpha$, where $\alpha$ represents the angular deviation in wind from the direction of the transect), is representative of the general trajectories of air masses in the PBL being advected from the Gulf of Mexico. While actual trajectories that do not follow straight paths may stay over emissions sources for long periods of time, large spatial deviations in wind direction will result in a reduction of the magnitude of $\bar{U} \cdot \cos \alpha$. Therefore, the time an air mass spends over an emissions source will be reflected in the magnitude of the resultant wind. In a well-mixed PBL, the alkane mixing ratios are assumed to be near constant with height, and the mixing ratio enhancement ($\bar{n}$) multiplied by the integrated molar density ($\int_{Z_0}^{Z_{PBL}} \rho(z) \, dz$) from the surface ($Z_0 = 122$ m at Floresville ) to the top of the PBL ($Z_{PBL}$) provides an estimate of the molar flux between the upwind and downwind locations. It is assumed that $\rho(z) = \rho_0 \cdot \exp\left(-\frac{z}{H}\right)$, where scale height $H = \frac{R_{air}T}{g}$, $R_{air} = 287$ J kg$^{-1}$ K$^{-1}$, $g = 9.81$ m s$^{-2}$, and the molar density of air at sea level $\rho_0 = 42.29$ mol m$^{-3}$ (*United States Committee on Extension to the Standard Atmosphere*, 1976). Lastly, a horizontal dimension ($\Delta x$) is necessary to produce an alkane flux for the region that is affecting the downwind receptor location. This was estimated as outlined in Sect. 2.4.

Meteorological data used in the mass balance approach were obtained from NOAA's North American Regional Reanalysis (NARR) (*Mesinger et al.*, 2005), a combined model and assimilated dataset with a horizontal resolution of approximately 32 km. Temperature and PBL height data for each date were obtained for the 3 hour period from 15:00 to 18:00 LST, representing general afternoon hours when the PBL is well mixed. Wind data were obtained for the previous 3 hour period of 12:00 to 15:00 LST when parcels were being advected over the EFS. Temperature and wind components at 950 mb were assumed to be representative of boundary layer conditions. Days with complicated meteorological conditions (e.g., precipitation, fronts, dry lines, or strong wind shear in the PBL) were discarded. The boundary layer heights from the NARR have been shown to have no strong bias compared to objectively determined PBL heights from sounding data at a site in Oklahoma (*Schmid and Niyogi*, 2012), although the correlation is moderate (as high as $R = 0.58$ in the winter and as low as $R = 0.39$ in the spring). While the use of the NARR introduces some uncertainty in the meteorological variables, we consider this to be the best available information for this data-sparse region where only surface observations are available. The uncertainty assigned to the meteorological variables is discussed in Section 2.5.

## 2.4   Horizontal dimension and production reference areas

The horizontal dimension in previous mass balance applications using aircraft data has typically come from an observation of background mixing ratios at the "edge" of the emissions plume (e.g. *Karion et al.*, 2015), where upwind and downwind mixing ratios become indistinguishable. Since the EFS can be considered a line source, but only one downwind measurement site is available, we defined the "edge" of the emissions plume using HYSPLIT's backward dispersion modeling tool in STILT mode

(*Hu et al.*, 2015). Model resolution was set to a 0.05° latitude × 0.05° longitude output grid (approximately 5 km resolution at these latitudes) using 12 km North American Mesoscale Model (NAM) meteorology input data. The model was set up to release 5,000 particles equally distributed in the PBL above the Floresville monitor site at 16:00 LST on each selected day using the estimated boundary layer depth from the NARR data. Particles were followed backwards for 20 hours and an integrated emissions impact map was created from particles entering the lowest layer (50 m agl). In almost all cases, the map was no longer changing after 8-14 hours of backward integration because all boundary layer particles had moved off-shore. Particle plots were used to further exclude days with significant wind shear in the boundary layer, as they do not fulfill the requirements for the mass balance technique.

The emissions impact map was assessed in two ways: (1) The near-field plume width was measured at the southern edge of the EFS as the representative horizontal measure necessary for the mass balance equation (Eq. 7) by assessing grid cell distances between the eastern and western edges of the plume. This choice was based on the assumption that alkane emissions are dominated by emissions in the EFS, that this width corresponds to the spread of back trajectory ensembles from the receptor location in Floresville, and that this dimension corresponds to the width of a plume under the uniform advection conditions necessary for mass balance had a continuous downwind measurement taken place for a source centered on the EFS. (2) The gridded map was overlaid with a map of natural gas and associated gas production for the thirty month period from July 2013 to December 2015, developed from county production data (*Railroad Commission of Texas*) equally distributed into the grid based on our assumption of a homogenously distributed source. All gas production in non-zero grid cells was accumulated to provide a reference number of upwind production potentially contributing to the measured downwind mixing ratios at the Floresville receptor. These numbers thus varied on a daily basis with wind direction and turbulence affecting the integrated impact map. Note that this estimate is based on the single receptor location, assuming it to be equivalent to an actual boundary layer "curtain" measurement undertaken via flying aircraft. Simulating the aircraft's "curtain" measurement via a particle release from numerous upwind locations would not substantially alter the result because of counter-acting consequences: A multi-point release throughout the downwind boundary layer would increase the width of the plume (impact map cross-section at southern EFS edge), increasing the total emissions estimate according to Eq. 7, but at the same time would also increase the production reference area where potential emissions occur. Thus our results would only change significantly if either upwind emissions or production were strongly non-homogenously distributed.

## 2.5 Monte Carlo Simulation

The errors arising from the variability and uncertainties of the alkane enhancement and the parameters derived from regional meteorology inputs were assessed using a Monte Carlo simulation, in which the emissions for each day were calculated one million times by randomly sampling the input parameters from either empirical or assumed probability distributions. A Monte Carlo simulation was performed for each day, allowing for the temporal variability and the dependence of the emissions on input variables to be assessed. The simulation was performed using the 'mc2d' package in R (*Pouillot and Delignette-Muller*, 2010). The absolute uncertainty in the afternoon alkane mixing ratios at each site associated with the precision of the instrument was represented by normal distributions about the afternoon alkane mixing ratios with relative standard deviations of 5.8%,

as discussed in Sect. 2.1. The compositions of four raw natural gas and four tank gas samples shown in Table 2 were used to produce normal distributions of the alkane ratios in raw natural gas and tank gas. The $u$ and $v$ components of the wind and the temperature were assigned normal distributions using the mean and standard deviation of the spatial variability in the NARR data over a 1° latitude by 1° longitude box situated in the central Texas coastal plain, with Floresville located at the northwest corner (Fig. 1). It is assumed that the meteorology in this region represents the general conditions to which air masses were subjected as they traveled inland from the Texas Coast towards Floresville.

There are several objective methods used to determine the PBL depth, and the uncertainty in the PBL depth has been shown to contribute to the uncertainty in previous mass balance measurements (e.g. *Karion et al.*, 2015). To the authors' knowledge, the uncertainty in the NARR PBL depth, which is estimated using the profile of turbulent kinetic energy (TKE), has not been quantified by the NARR maintenance team (*Mesinger et al.*, 2005). However, the NARR has been compared to PBL depths estimated from radiosonde data. *Lee and De Wekker* (2016) found that objectively analyzed PBL depths using radiosonde data in Virginia differed from PBL depths estimated using a bulk Richardson method with the NARR data. The standard deviation of the difference between the two methods was 453 m when averaged over one year and the NARR PBL depths exhibited a high bias of 157 m when compared to the radiosonde PBL depths. A similar study at a site in Oklahoma (*Schmid and Niyogi*, 2012) examined the difference between objectively analyzed PBL depths using radiosonde data and the standard NARR PBL depths using the TKE method. The correlation between the NARR and the radiosonde PBL depths were slightly lower in this study when compared to *Lee and De Wekker* (2016). To be consistent with these authors' findings, we have assumed that the uncertainty in the NARR PBL depth may be represented by a standard deviation of 500 m, which is an average of 28% of the NARR PBL depth over the days used in the study (Section 3.2). Therefore, the PBL depths are represented in the Monte Carlo error estimate as a normal distribution centered on the average of the NARR PBL depths over the 1° latitude by 1° longitude box and a standard deviation of 500 m.

## 3 Results and Discussion

### 3.1 Ethane trends

*Schade and Roest* (2015) briefly discussed long-term trends in ethane mixing ratios at TCEQ sites around the EFS, and an update is provided in Fig. 2. Here, we present the results from Kruskal-Wallis rank-sum and Dunns tests (*Kruskal and Wallis*, 1952; *Dunn*, 1964) performed on ethane mixing ratios vs. year. At the Corpus Christi – Hillcrest site, no set of years exhibited statistically significant ($p < 0.05$) differences in ethane mixing ratios under southeasterly wind regimes. At Clute, ethane was statistically significantly higher ($p < 0.05$) in 2015 than it was in 2007-2011, though no other years showed significant differences. We attribute recent increases in ethane mixing ratios in Clute to changes in emissions from local point sources, as the neighboring city of Freeport is a hub of petroleum processing and transportation (*Bonney*, 2014; *Ryan*, 2014). The data suggest that background ethane levels over the Gulf of Mexico did not significantly change during the development of the EFS. However, ethane mixing ratios at the San Antonio site are statistically significantly higher ($p < 0.05$) in later years than in earlier years. Ethane was higher in 2011 than 2007-2009, higher in 2012 than 2007-2011, and higher in 2013-2015 than

in 2007-2010. The Floresville site was not installed until 2013 so the long term trend in ethane mixing ratios at that location is unknown. However, Floresville observed the highest ethane mixing ratios in the region from 2013-2015. While there is no evidence that ethane mixing ratios along the coast increased over time during southeasterly flow, ethane did increase downwind of the EFS during its development.

## 3.2 Alkane Enhancement

During the thirty-month period from July 2013 through December 2015, a total of 69 days were found to have 3 out of 4 trajectories identified as southeasterly, appropriate meteorological conditions, and 75% completeness at both Floresville and Oak Park. One of these days (18 March 2015) had alkane enhancement values that were outliers. Since we cannot exclude that the downwind measurement site of Floresville was influenced by a plume on this date, it was not considered for analysis. The majority of the remaining 68 dates, which occured between August 2013 through August 2015, fell into the summer and fall months, when southerly and southeasterly flow are commonplace in south-central and coastal Texas (*Texas Commission on Environmental Quality*, 2015). Supporting Table S2 shows the observed alkane enhancements for each day, along with the alkane emission estimates (Sect. 3.3) and meteorology. The median alkane enhancements for the set of 68 days were as follows: ethane – 2.4 ppb with an interquartile range (IQR) of 2.0-3.1 ppb; propane – 1.9 (IQR of 1.4-2.5) ppb; *n*-butane – 0.8 (0.6-1.1) ppb; and isobutane – 0.4 (0.3-0.5) ppb. All observed alkane enhancements were positive.

Supporting Fig. S2 shows a timeline of the afternoon ethane mixing ratios at both Oak Park and Floresville for the set of 68 days with southeasterly flow. Ethane mixing ratios during the warm season (summer and fall) were generally low at both Oak Park and Floresville and higher at the two sites during the cool season (winter and spring). This seasonal variability conforms with current understanding of temporal hydrocarbon variability (*Helmig et al.*, 2016). The enhanced photochemical oxidation of ethane during the summer months explains the low background ethane observed in the onshore flow in Oak Park. (*Haman et al.*, 2012).

## 3.3 Partitioning of alkane sources

The enhancements of propane, butanes, and pentanes were highly correlated with ethane enhancements between Oak Park and Floresville, suggesting a co-emission from sources of natural gas. The strongest correlation was observed between ethane and propane (Fig. 3). Pentanes (and to a lesser extent, butanes) may be impacted by emissions from automotive traffic (*Tsai et al.*, 2006; *Ho et al.*, 2009; *Simpson et al.*, 2012) and chemistry, as pentanes have atmospheric lifetimes of 1.5 days (298 K, $[OH] = 2.0 \times 10^6$) (*Atkinson and Arey*, 2003). Therefore, pentanes were not used to partition emissions from raw natural gas and tank gas sources and an emission rate was not calculated. Nonetheless, the isopentane to n-pentane enhancement ratio between Oak Park and Floresville (supporting Fig. S3), which will remain close to constant despite chemistry, was 1.17 ($p < 0.001$), indicating that alkane emissions are largely influenced by oil and gas (*Gilman et al.*, 2013; *Swarthout et al.*, 2016). By comparison, the same ratio at the Old Highway 90 site in San Antonio was 2.25, which falls within the bounds of the traffic-emission driven urban pentane ratio. For these reasons, the enhancements of short-chain alkanes during advection over the EFS are most likely dominated by emissions from oil and gas production. Note that the alkane enhancement distributions

(Table S2) are skewed, with the upper bounds possibly representing the influence of individual plumes from UOG exploration activities.

Propane-to-ethane ratios were used to partition emissions from raw natural gas and tank gas sources. Within each Monte Carlo simulation, the fraction of the ethane enhancement from raw natural gas sources varied largely, with the 95% confidence interval often bounded by physically unreasonable numbers. This was due to the large uncertainty of the propane-to-ethane ratios in raw natural gas and tank gas samples. However, the median fraction for each day showed less variability, with a median of 45% (IQR of 34-52%) of ethane attributed to raw natural gas sources. The raw natural gas source estimate was significantly more constrained due to the availability of raw gas composition data, while no tank gas composition data were available for the EFS. The median methane enhancement estimate for all 68 days was 8.9 (IQR of 6.8-12.4) ppb, of which 83% (IQR of 76-86%) was attributed to raw natural gas sources. Higher alkanes were also partitioned, and their relative contributions from raw natural gas sources over the 68 days were: propane – 18% (IQR of 11-22%); $n$-butane – 10% (7-13%); and isobutane – 17% (11-22%). While the majority of methane emissions were due to emissions of raw natural gas, liquid storage tanks dominated the emissions of these higher alkanes

The raw natural gas source fractions based on $n$-butane-to-ethane and isobutane-to-ethane ratios often did not match that of propane-to-ethane for individual days, although their probability distribution functions over all days overlapped (supporting Fig. S4), suggesting that the partitioning is consistent and reasonable when integrated over multiple sources in a larger region. It should be noted, however, that Floresville is located closer to the oil producing window of the EFS than the gas producing window and the influence of raw natural gas production may be somewhat diminished due to this distance. Nonetheless, the contribution of tank gas sources to observed alkane enhancement ratios suggests that liquid storage tanks are a very important source of alkane emissions between Oak Park and Floresville, a result that agrees with hydrocarbon emission studies in many other liquids-rich U.S. shale plays.

### 3.4  Mass balance results

### 3.4.1  Dispersion plumes and upwind production areas

Figure 4 shows a representative backward dispersion plume output overlaid on a gridded production map to illustrate the relative emissions estimate. Emission plumes for other days varied in width, but generally overlapped the same counties. Since the EFS acts as a line source, all overlapping grid cells with non-zero production numbers were weighed equally. This area includes portions of several counties in the EFS – notably Atascosa, Bee, Karnes, Live Oak, and Wilson counties – which are members of AACOG (*Alamo Area Council of Governments (AACOG)*, 2015). *Pacsi et al.* (2015) found that, among all counties in the EFS, $NO_x$ emissions from these counties had the greatest impact on ozone enhancement in Bexar County, home to the city of San Antonio. For these reasons, quantifying emissions from these counties in the core of the EFS is a particularly important step in assessing the air quality impacts of oil and gas operations in the San Antonio area. While production was considered over all grid cells overlapped by the backward dispersion plume, counties in the EFS dominate regional production and are likely responsible for the observed alkane enhancements.

### 3.4.2 Emission estimates

The alkane emissions estimates from the upwind production regions were estimated using a Monte Carlo simulation for each day, and the distributions of the median emission rates were found as follows: methane – 189 (IQR of 136-376) $\times 10^3$ kg day$^{-1}$; ethane – 97 (68-168) $\times 10^3$ kg day$^{-1}$; propane – 109 (75-185 ) $\times 10^3$ kg day$^{-1}$; *n*-butane – 64 (46-104) $\times 10^3$ kg day$^{-1}$; and isobutane – 27 (18-52) $\times 10^3$ kg day$^{-1}$. In comparison, VOC emissions estimated by AACOG (not including ethane) for the set of central EFS counties from southeast to southwest of Floresville were only 88 $\times 10^3$ kg day$^{-1}$ for calendar year 2012 (*Alamo Area Council of Governments (AACOG)*, 2013b), while total EFS VOC emissions were estimated at 203 $\times 10^3$ kg day$^{-1}$. This suggest that EFS VOC emissions used in ozone modeling may be underestimated by at least a factor of two, likely more.

To estimate relative methane losses, the raw natural gas-only and the total mass emission of methane were converted into a volume of natural gas using the ideal gas law at standard temperature and pressure for natural gas volume reporting (*Texas Statutes*, 1977) and a natural gas methane content based on available raw natural gas data (Table 2). The volume of the emitted natural gas was then compared to the produced natural gas at gas wells and associated gas at oil wells in the production reference area outlined in Sect. 2.4 and 3.4.1. Emission rates for individual days were highly uncertain, with the bounds of the 95% confidence interval often spanning an order of magnitude (supporting Fig. S5). However, median emission rates over all days were less variable, with a median total emission rate of 1.0% (IQR of 0.7-1.6%) and a raw natural gas-only emission rate of 0.7% (0.5-1.3%). While the EPA's estimated emission rate of 1.6% falls within the bounds of our total emission estimate, our median emission rate is lower than that of the EPA. The emission rate displayed no significant trend over the 2013-2015 time period, and its correlations with independent meteorological variables and ethane enhancement over time were weak (supporting Fig. S6 and S7), which is to be expected for a continuous anthropogenic emission source. However, the uncertainty in the emission rate within each Monte Carlo simulation showed a strong dependence on the uncertainty of the ethane content in raw natural gas samples – a variable with large relative uncertainty that was used to estimate methane emissions – and, to a lesser extent, meteorology (supporting Fig. S8). We find that the lack of data regarding the composition of both raw natural gas and vented gases from liquid storage tanks impedes a higher precision top-down emission rate estimate. Nonetheless, repeated emission estimates over a large set of days show consistent and reasonable emission rates that can be attributed to UOG operations.

## 4 Conclusions

Our study used ethane as a tracer for alkane emissions from UOG emissions for a region in southern Texas, where oil and gas production is dominated by the core of the EFS. Data from the TCEQ show that alkane mixing ratios downwind from the shale have increased in tandem with oil and gas production rates in the EFS. This trend, along with the strong correlation of ethane enhancements with both propane and butane enhancements, show that emissions from UOG production in the EFS are responsible for the observed alkane enhancements across the shale. Using a mass balance approach and a Monte Carlo error estimation, we calculate ethane emissions of 97 (IQR of 68-168) $\times 10^3$ kg day$^{-1}$ from areas between the Texas Coast

and the downwind receptor site in Floresville for a set of 68 days from August 2013 through August 2015. Using typical ethane-to-methane ratios (an average of 0.14), we estimate methane emissions of 189 (136-376) $\times$ $10^3$ kg day$^{-1}$. These emissions represent 1.0% (0.7-1.6%) of the produced natural gas – including associated gas at oil wells – in the region upwind of Floresville. We show through the partitioning of raw natural gas and tank gas emissions that raw natural gas sources account

for three quarters of all methane emissions, with a raw natural gas-only relative emission rate of 0.7% (0.5-1.3%). Note that these emission rates are expressed as a fraction of produced natural gas as opposed to produced energy. In liquids-rich shale plays such as the EFS, expressing emission rates as a fraction of produced energy may be a more appropriate measure of emissions, especially when comparing energy losses to other sources (e.g. coal). However, our findings suggest that energy losses in the form of VOC emissions may be an important consideration when estimating energy losses from liquids-rich shale

plays.

We find that tank gas sources account for more than half of higher alkane emissions – notably 90% of *n*-butane emissions. Since the petroleum production in this region is dominated by counties within the EFS, we conclude that UOG activities in the shale area are largely responsible for these emissions. Our natural gas emission rate estimate falls within the bounds of other top-down studies (e.g. *Peischl et al.*, 2015) and overlaps the EPA's most recent methane emissions estimates from its 2016

greenhouse gas inventory (*U.S. Environmental Protection Agency*, 2016). However, our median total emission rate including tank gas sources is lower than the EPA's emission rate. This suggests that methane emissions in the EFS might be lower than current average nationwide emission rates in bottom-up inventories. However, NMVOCs co-emitted with methane are likely underreported and underestimated in inventories used for air quality modeling studies in southern Texas. The partitioning of emissions from raw natural gas sources and liquid storage tanks confirms that tank gas is an important source of short-chain

alkane emissions in the EFS, with the enhancement of propane during air mass transport over the EFS nearly as large as that of ethane. Furthermore, a recent assessment of hemispheric short-chain hydrocarbon emission trends highlighted an unknown source with a methane/ethane ratio that is lower than that of RNG (*Helmig et al.*, 2016). Existing data for raw natural gas and tank gas compositions show that the typical methane/ethane ratio in tank gas emissions are relatively low compared to that of raw natural gas emissions. Our results show that emissions of alkanes from liquid storage tanks account for 17% of methane,

55% of ethane, 82% of propane, 90% of *n*-butane, and 83% of isobutane emissions from the EFS. These emissions are likely to contribute to the unknown NMHC source identified by *Helmig et al.* (2016).

While alkanes have been shown to dominate the OH reactivity at Floresville, the scarcity of trace gas measurements within the EFS prevents more thorough NMVOC emissions estimates. Our calculations indicate that propane and butanes emissions alone exceed current inventory numbers for the EFS by approximately a factor of two, which can have significant impacts

on ozone modeling, particularly if NO$_x$ emissions are underestimated as well. Hence, we stress the need for increased spatial coverage of VOC, NOx, and greenhouse gas monitoring in and around the shale area to improve upon existing emission inventories. Such improvements are needed before the air quality impacts of the EFS can be accurately quantified. As the unconventional oil and gas industry in the EFS continues to grow, the climate and air quality impacts associated with emissions from the shale need to be addressed. This is especially true as the San Antonio metropolitan area may be designated as a

nonattainment area by the EPA. However, existing emissions estimates are uncertain and variable, and need to be improved before the impacts on air quality can be quantified.

*Author contributions.* G. Roest gathered and analyzed data, performed the emissions estimate, and prepared the manuscript. G. Schade provided input for the emissions estimate, performed the plume dispersion analyses and plume width estimate, and assisted in manuscript preparation.

*Competing interests.* The authors declare that they have no conflict of interest.

*Acknowledgements.* We would like to thank TCEQ for providing automated GC data in the summer and fall of 2014, which are now available online at http://www17.tceq.texas.gov/tamis/. Funding was provided internally within the Department of Atmospheric Sciences at Texas A&M University.

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

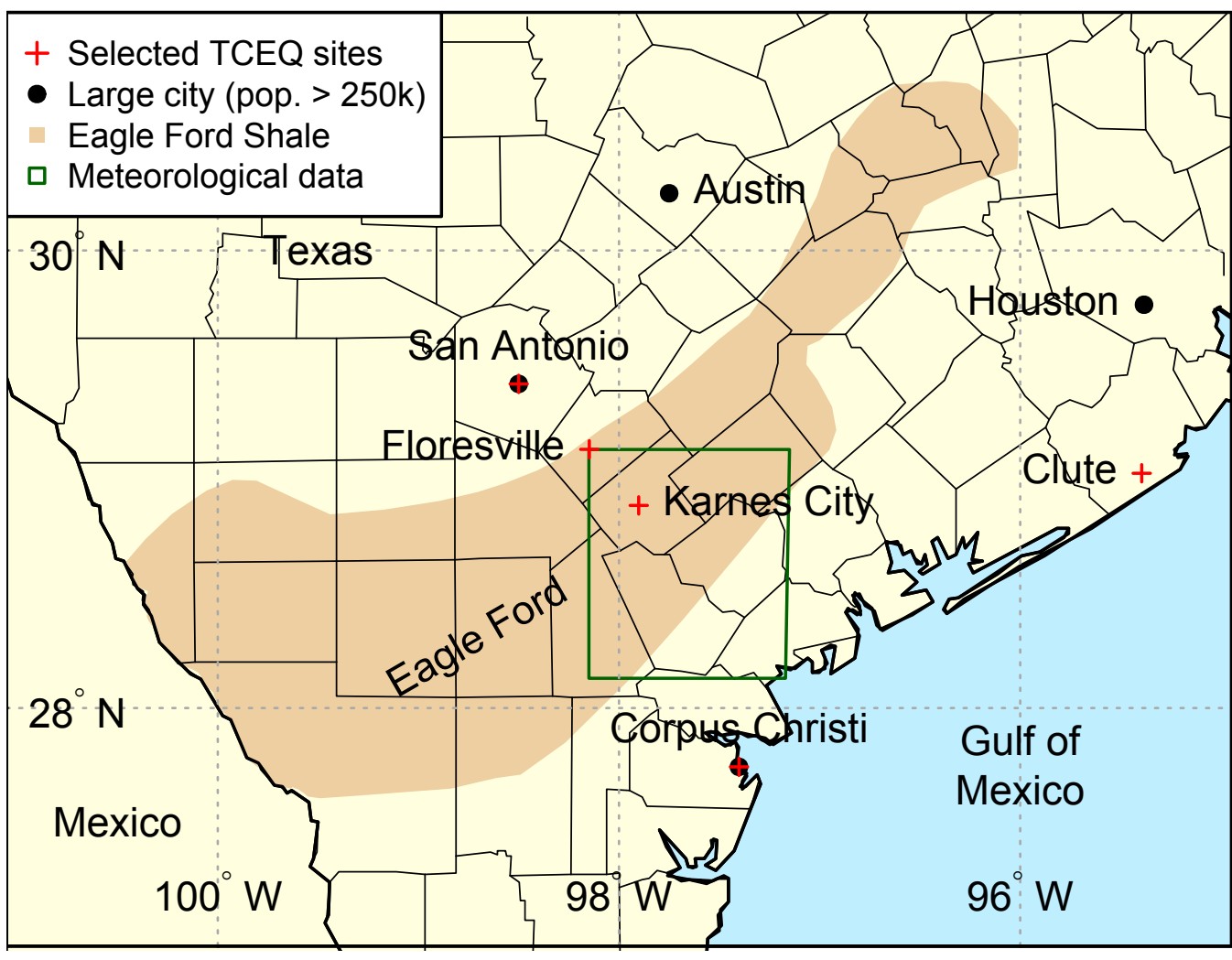

**Figure 1.** Selected TCEQ NMVOC monitoring sites and large cities near the Eagle Ford. The green box shows the $1°$ latitude by $1°$ longitude box in which the meteorology was assessed.

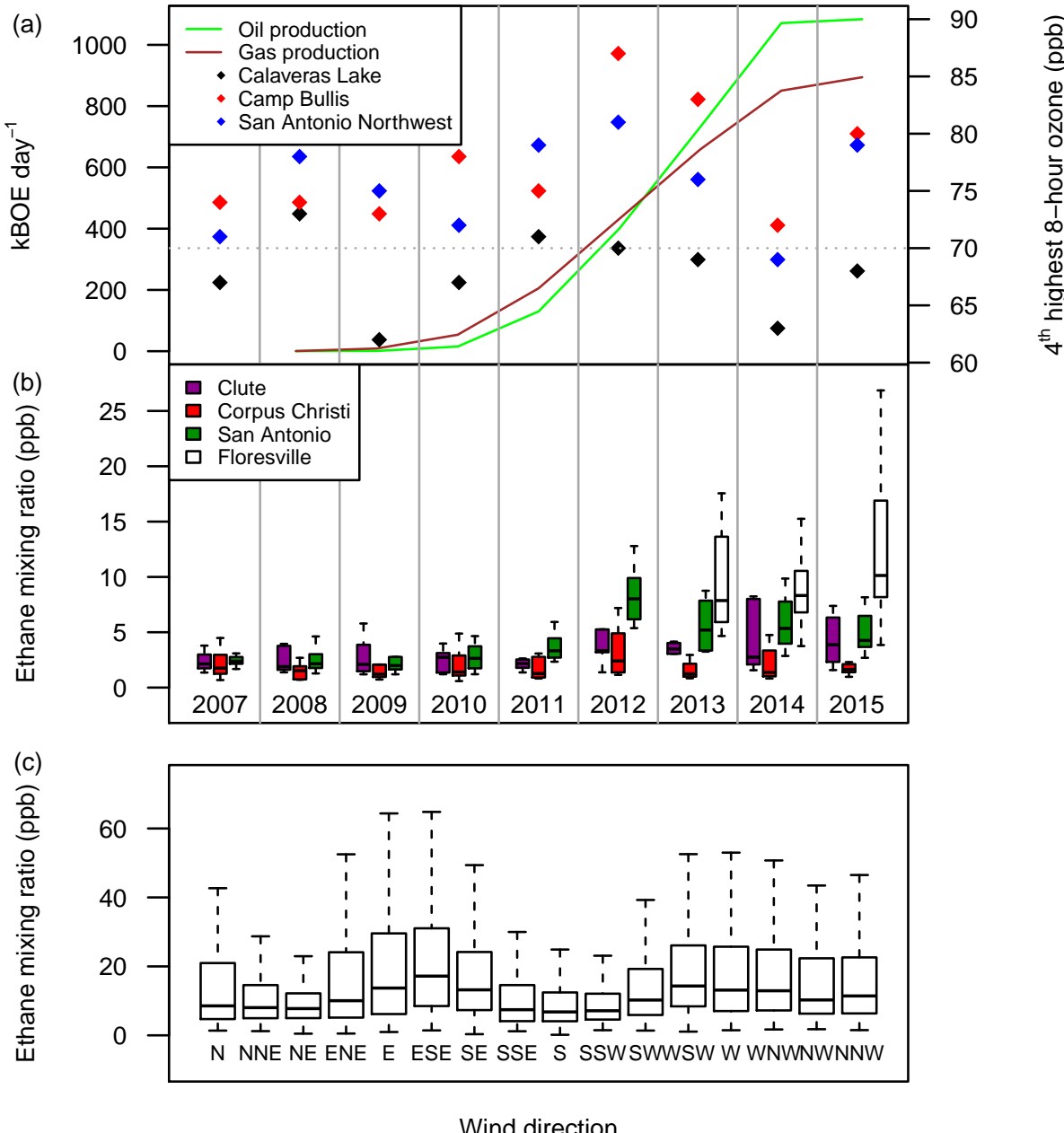

**Figure 2.** Adapted version of Fig. 2 in *Schade and Roest* (2015), updated to include data through 2015. (a) Oil and gas production rates in the Eagle Ford and 4th highest maximum 8-hour ozone values at 3 sites in San Antonio. (b) Timeline of 24-hour ethane mixing ratios at 4 sites near the Eagle Ford Shale. Days were used only if 3 out of 4 back-trajectories originating from San Antonio – Old Highway 90 were binned as southeasterly. Data at Floresville begins in July of 2013. (c) Ethane mixing ratios vs. wind direction at Floresville, with elevated mixing ratios under E to SE or SW to W winds, when trajectories would generally allow for the accumulation of emissions as winds have a component that is parallel to the shale axis. Ethane is also elevated under NW winds, likely due to higher ethane in continental air masses and local emissions from the San Antonio metropolitan area.

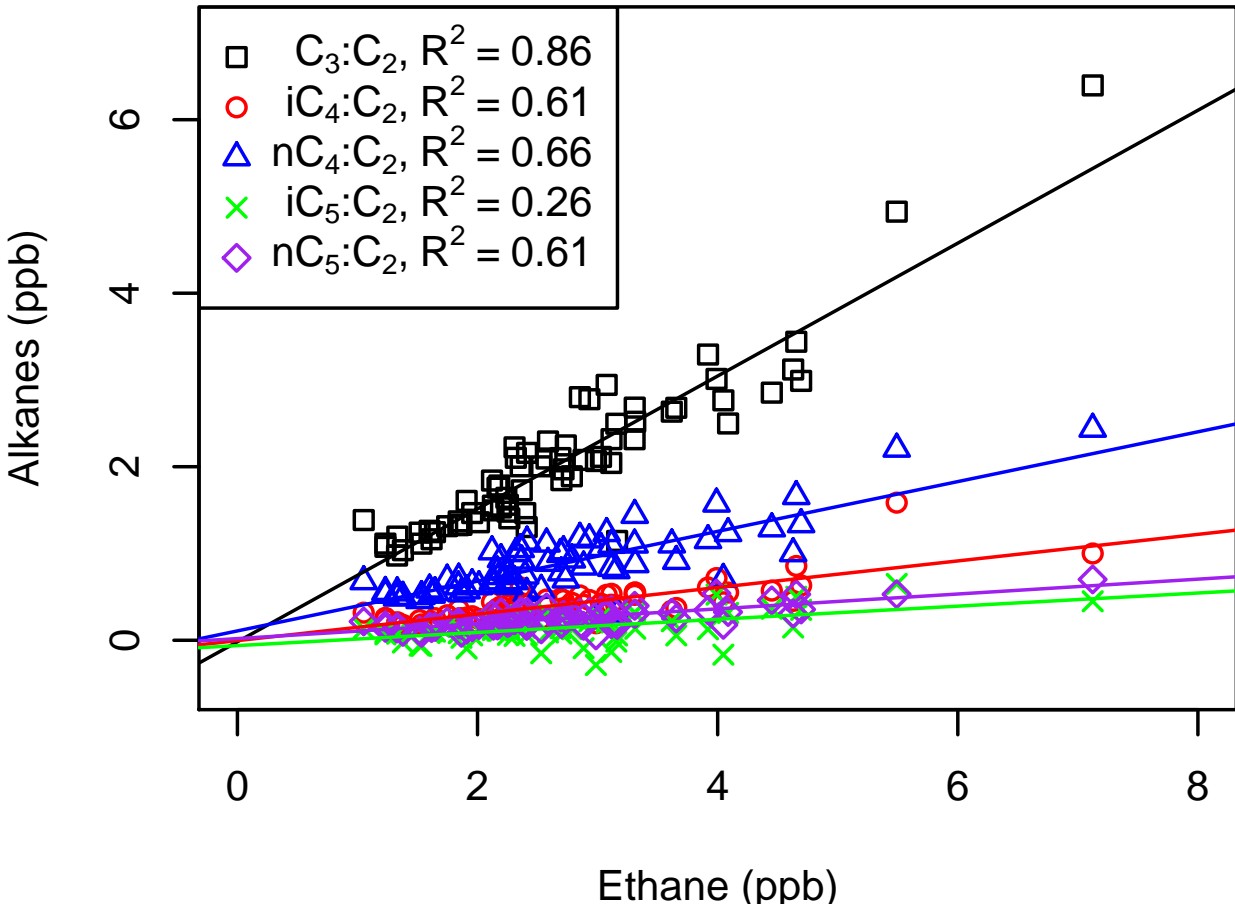

**Figure 3.** Correlations of propane, butane, and pentane enhancements with ethane enhancement. All correlations were highly statistically significant ($p < 0.001$). While ethane and propane showed the strongest correlation, all alkanes showed a positive correlation with ethane.

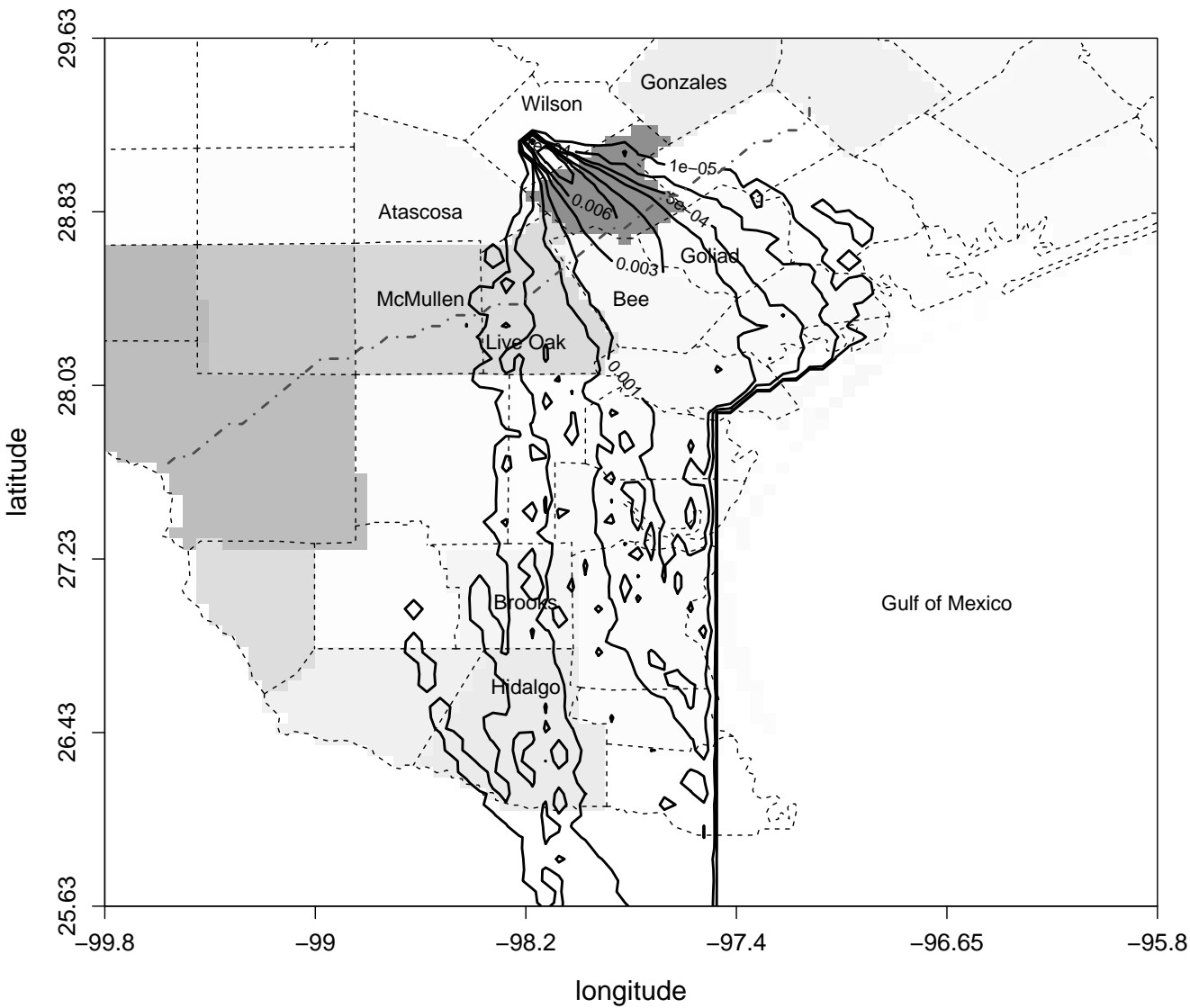

**Figure 4.** An example of an integral backward dispersion plume map created from 5000 particles released above the Floresville receptor site and followed backwards in time (20 hours) into the model lowest vertical level (<50 m agl) to assess surface emitter impacts. The raw map was normalized to its total after removing all surface impacts over the Gulf of Mexico. The grey shading underlying the plume map identifies counties with the sum of natural gas and associated gas production, darker shading indicating higher production rates. The darkest shading indicates Karnes County, not labeled for clarity. Many counties, including Wilson County where Floresville is located, are lightly shaded due to the relatively low production of natural gas. The dark grey, jagged line extending from the west-southwest near the Mexican border to the east-northeast south of Gonzales County marks the southern "edge" of the EFS.

**Table 1.** Description of TCEQ sites used in this study.

| Site Name | AQS Code | Lat ($^{\circ}$N) | Lon ($^{\circ}$W) | Sample Duration | Samples Collected | Use in Study |
|---|---|---|---|---|---|---|
| Clute | 480391003 | 29.01 | 95.40 | 24 hour | Once every 6 days | Long-term trends |
| Corpus Christi – Hillcrest | 483550029 | 27.81 | 97.42 | 24 hour | Once every 6 days | Long-term trends |
| Corpus Christi – Oak Park | 483550035 | 27.80 | 97.43 | 40 min | Hourly, automated | Emission estimate |
| Floresville Hospital Boulevard | 484931038 | 29.13 | 98.15 | 40 min | Hourly, automated | Emission estimate |
| San Antonio – Old Highway 90 | 480290677 | 29.42 | 98.58 | 24 hour | Once every 6 days | Long-term trends |

**Table 2.** Ethane content in raw natural gas and tank gas samples by mol percent and associated ethane/alkane ratios

| Ratio | $C_2$ (mol %) | $C_2/C_1$ | $C_2/C_3$ | $C_2/iC_4$ | $C_2/nC_4$ | $C_2/(iC_5 + nC_5)$ |
|---|---|---|---|---|---|---|
| | $4.51^a$ | 0.05 | 2.20 | 9.40 | 8.84 | 11.00 |
| | $9.15^b$ | 0.11 | 2.97 | 8.55 | 9.24 | 9.63 |
| Raw Natural Gas | $13.20^b$ | 0.17 | 2.63 | 12.34 | 10.08 | 18.86 |
| | $15.88^b$ | 0.22 | 2.55 | 36.93 | 12.70 | 31.76 |
| Mean | 10.69 | 0.14 | 2.59 | 16.80 | 10.22 | 17.79 |
| | $13.07^c$ | 0.84 | 0.75 | 2.52 | 1.08 | 0.47 |
| | $16.83^c$ | 0.72 | 1.13 | 3.21 | 1.59 | 0.78 |
| Tank Gas | $14.04^c$ | 0.61 | 0.89 | 3.32 | 1.45 | 0.65 |
| | $13.58^c$ | 0.47 | 0.96 | 4.02 | 1.54 | 0.64 |
| Mean | 13.48 | 0.66 | 0.93 | 3.27 | 1.42 | 0.64 |

[a] (*Pring*, 2012), [b] (*Todd*, 2011), [c] (*ENVIRON International Corporation*, 2010)