# Peer review of "Quantifying alkane emissions in the Eagle Ford Shale using boundary layer enhancement"

_Atmospheric Chemistry and Physics, 2016_

## Referee Comment (RC1) · Anonymous Referee #1 · 15 Dec 2016

In the manuscript "Quantifying alkane emissions in the Eagle Ford Shale using boundary layer enhancement", Roest and Schade use atmospheric enhancement of alkanes from five measurements sites, with back trajectory and dispersion modeling, to quantify alkane and methane emissions in the Eagle Ford Shale region in Texas. They determine that an emission rate for raw natural gas that is consistent with the EPA's methane bottom-up estimate (1.1% of production). When they also consider storage tank leakage, the median emission rate is, however, 2.2% of production. Knowledge of shale gas regional emissions in general, is limited, and this very well written paper contributes significantly to the scientific base of knowledge. The authors are careful to adequately describe assumptions in their work (associated with mass balance approach and ethane/alkane ratios) and also include a Monte Carlo simulation to assess uncertainty. I recommend this paper for publication in Atmospheric Chemistry and
Physics, with technical/minor revisions, listed below.

Specific comments

Section 2.1: I had a hard time figuring out how many sites were used in the analysis based on the description in Section 2.1 and Figure 1. Table S1 was helpful for me to understand - consider moving it to the main paper. Indicate that there are five total sites. Section 2.1 says there are "several sites in Corpus Christi, including Hillcrest and Oak Park" but only those two are indicated in Table S1. I would also use a different color and/or more prominent symbol for the TCEQ sites in Figure 1 - as it is the symbols are hard to find, especially ones inside the circles indicating large cities.

Section 3.1: Mention Floresville. Even though the measurements didn't start until 2013, the signal is prominent in the figure.

page 11, line 12: Specify that the emission rate displayed no trend *over the period 2013-2015*. (Otherwise seems inconsistent with results from Figure 2.)

There are many acronyms in this paper. Readability might be improved if the authors wrote some of them out instead. RNG (raw natural gas) and TG (tank gas), for example.

Technical comments

page 2, line 7: "associated with gas produced" instead of "associated gas produced"

page 10, line 1: "significantly more constrained" instead of "significantly more constraint"

---

## Referee Comment (RC2) · Anonymous Referee #2 · 29 Dec 2016

Roest and Schade use alkane data from ground sites in and around the Eagle Ford shale in Texas, along with a transport model, to determine methane and other small chain alkane emission rates for the region. The ground site data come from several sites upwind and downwind of the Eagle Ford shale. Based on estimates of wind fields, planetary boundary layer height, and emission locations, the authors perform a mass balance analysis to estimate emissions from the Eagle Ford oil and gas production. They conclude that inventory emissions levels are low by about a factor of two, which could affect the accuracy of air quality models.

**General comments**

This paper provides an emission estimate from a region with extensive oil and gas production whose emissions are not well known, and would therefore be an important

addition to the body of knowledge concerning methane and alkane emissions from oil and gas production regions in the U.S. However, I have some concerns regarding the analysis that I think must be addressed before this paper is ready for publication. My main concerns are discussed in the next two paragraphs. Some lesser concerns are brought up in the Other Comments section.

I am concerned with using Barnett tank alkane ratios to represent tank emissions from the Eagle Ford. First, the alkane ratios could be significantly different from the two regions. My understanding of the Eagle Ford shale is that produces a very wet mixture of hydrocarbons. Do the authors have data from any other oil producing regions in the U.S., like the Bakken or a traditional oil producing region? If they used those ratios, how would that affect the analysis results? I think more work will need to be done to show the effects of this assumption, especially since it plays such a large role in the results.

Another concern I have is with the use of ground-based sampling to represent the entire vertical extent of the planetary boundary layer. I think some discussion of the location of possible sources of methane relative to the sampling sites is necessary. This is especially true for the Floresville site, which may be influenced by emissions that have not mixed completely through the planetary boundary layer on more days than just 18 March 2015.

**Other comments**

p. 1, Line 25, is carbon monoxide a HAP? I don't see it here: https://www.epa.gov/haps/initial-list-hazardous-air-pollutants-modifications

p. 2, Line 10, please add Olaguer et al. to the References

p. 2, Line 14, How does the 5750 Gg of methane compare to the EPA GHG inventory? If they are the same, I'd cite the GHG inventory. If they are different, I'd still use the GHG inventory, but note the difference. Also, does this number include emissions from
petroleum production as well as natural gas? Since you included associated gas in the leak rate calculation for the Eagle Ford on p. 11, line 27, the national leak rate should also include emissions from petroleum production.

p. 3, Line 16, This emission rate is somewhat misleading. Schneising et al. reported an energy content leak rate, which is not the same as a natural gas leak rate. The energy content leak rate takes into account the oil produced as well as the natural gas. See Howarth [Energy and Emission Control Technologies, 2015, p. 48] or Peischl et al. [JGR-Atmospheres, 2016, p. 2] for a discussion on this issue.

p. 4, Line 21, Why did you use the EDAS 40 km dataset for meteorology over others? Perhaps include a sentence explaining your choice.

p. 5, Line 1, Please show a time series of the background upwind mixing ratios and the enhanced mixing ratios at Floresville. This will give the reader a sense of how well the background sites represent the background air impacting the Floresville site.

p. 7, Line 7, Please provide some discussion of the PBL height and what effect the uncertainty of the modeled PBL height has on the analysis. Has the modeled PBL height been verified using LIDAR or aircraft measurements?

p. 9, Line 1, Did you not see a seasonal change in background ethane due to greater chemical loss during the summer?

p. 11, Line 33, A comparison with the EPA inventory estimate from petroleum production would be a fairer one, considering how much oil is produced in the Eagle Ford shale.

Conclusions section, Please include a time frame for your emissions estimates. Are they for the entire study period? If so, please state this explicitly in the Conclusions.

Grammar suggestions

p. 3, Lines 7-10, This is a long sentence. Consider splitting it up into two.
p. 10, Line 1, I'm not sure "constraint" is a verb, unless it is an old-timey past tense.

---

## Author Response (AR1)

The authors' responses to Anonymous Referee #1 are below. Each response is provided below the Referee's original comments.

Specific comments:

"Section 2.1: I had a hard time figuring out how many sites were used in the analysis based on the description in Section 2.1 and Figure 1. Table S1 was helpful for me to understand - consider moving it to the main paper. Indicate that there are five total sites. Section 2.1 says there are "several sites in Corpus Christi, including Hillcrest and Oak Park" but only those two are indicated in Table S1. I would also use a different color and/or more prominent symbol for the TCEQ sites in Figure 1 - as it is the symbols are hard to find, especially ones inside the circles indicating large cities."

[Figure]

Response: Table S1 has been moved to the main text as Table 1 and the selection of the two sites in Corpus Christi has been justified. Figure 1 has also been updated and is attached.

"Section 3.1: Mention Floresville. Even though the measurements didn't start until 2013, the signal is prominent in the figure."

Response: The elevated ethane mixing ratios from 2013-2015 have been added to the discussion.

"page 11, line 12: Specify that the emission rate displayed no trend *over the period 2013-2015*. (Otherwise seems inconsistent with results from Figure 2.)"

Response: This comment has been accepted and addressed in the text.

"There are many acronyms in this paper. Readability might be improved if the authors wrote some of them out instead. RNG (raw natural gas) and TG (tank gas), for example."

Response: Except for subscripts in equations, RNG and TG have been replaced with "raw natural gas" and "tank gas".

Technical comments:

"page 2, line 7: "associated with gas produced" instead of "associated gas produced""

Response: "Associated gas" is often used to refer to natural gas that is coproduced at oil wells. This change was not accepted. Instead, "associated gas" was italicized to indicate that it is a phrase.

"page 10, line 1: "significantly more constrained" instead of "significantly more constraint""

Response: This comment has been accepted and addressed in the text.

[Figure]

Legend:
+ Selected TCEQ sites
● Large city (pop. > 250k)
▪ Eagle Ford Shale
□ Meteorological data

30° N    Texas

● Austin

San Antonio

Houston ●

Floresville +

+ Karnes City

Clute +

Eagle Ford

28° N

Corpus Christi

Gulf of
Mexico

Mexico

100° W          98° W          96° W

**Fig. 1.** Figure 1 in the manuscript has been updated to address the Referee's first comment.

[Figure]

Atmos. Chem. Phys. Discuss.,
doi:10.5194/acp-2016-861-AC2, 2017

[Figure]

The authors' responses to Anonymous Referee #2 are below. Each response is provided below the Referee's original comments.

General comments

"This paper provides an emission estimate from a region with extensive oil and gas production whose emissions are not well known, and would therefore be an important addition to the body of knowledge concerning methane and alkane emissions from oil and gas production regions in the U.S. However, I have some concerns regarding the analysis that I think must be addressed before this paper is ready for publication. My main concerns are discussed in the next two paragraphs. Some lesser concerns are brought up in the Other Comments section."

[Figure]

"I am concerned with using Barnett tank alkane ratios to represent tank emissions from the Eagle Ford. First, the alkane ratios could be significantly different from the two regions. My understanding of the Eagle Ford shale is that produces a very wet mixture of hydrocarbons. Do the authors have data from any other oil producing regions in the U.S., like the Bakken or a traditional oil producing region? If they used those ratios, how would that affect the analysis results? I think more work will need to be done to show the effects of this assumption, especially since it plays such a large role in the results."

Response: We agree that using alkane ratios from liquid storage tank emission samples from the Eagle Ford Shale would be the most appropriate data to use in this study. However, we are not aware of such data publicly available for the Eagle Ford. While we have inquired about non-public data with two possible sources, our requests were unsuccessful. The available data, i.e. the sampled emissions from liquid storage tanks in the Barnett Shale are variable in composition and this is incorporated into our Monte-Carlo error analysis. The composition of emissions from oil and condensate storage tanks in other areas of Texas (Hendler et al., 2009) are also largely variable. We assume that the average composition of liquid storage tank emissions in the Eagle Ford falls within that variability of the Barnett Shale samples, although this assumption does introduce an uncertainty in our analysis. The text has been updated to emphasize this assumption and the associated uncertainty.

"Another concern I have is with the use of ground-based sampling to represent the entire vertical extent of the planetary boundary layer. I think some discussion of the location of possible sources of methane relative to the sampling sites is necessary. This is especially true for the Floresville site, which may be influenced by emissions that have not mixed completely through the planetary boundary layer on more days than just 18 March 2015."

Response: We agree that vertical measurements of alkane mixing ratios would validate the assumption that upwind emissions have been mixed through the planetary bound-

ary layer (PBL). An aircraft campaign in the Barnett Shale has shown that methane emissions were thoroughly mixed downwind [Karion et al., 2015]. While we do not have alkane measurements in the vertical, the assumption of well-mixed upwind emissions is defensible. Our analysis shows that the long-term increasing trend in ethane enhancement between Corpus Christi and San Antonio parallels the development of the Eagle Ford Shale, suggesting that the Eagle Ford is responsible for the emissions that have led to said increase. Furthermore, the Eagle Ford Shale is sufficiently upwind of Floresville such that a discrete plume from a nearby source should be thoroughly dispersed in the afternoon PBL before it reaches Floresville. This is based on the finding that the nearest potential source is approximately 12 km upwind (a well pad south of the site), while all other sources are at least 15 or more kilometers upwind. Under typically conditions used in this study with boundary layers of 1.5-2 km depth and convective velocity scale values of 1-2 m s-1, vertical dispersion occurs within approximately 30 minutes, while horizontal transport at typical wind speeds of 5 m s-1 will require more than 30 min to reach the Floresville site.

Other comments

"p. 1, Line 25, is carbon monoxide a HAP? I don't see it here: https://www.epa.gov/haps/initial-list-hazardous-air-pollutants-modifications"

Response: This has been addressed in the text.

"p. 2, Line 10, please add Olaguer et al. to the References"

Response: Olaguer (2012) is already included in the references.

"p. 2, Line 14, How does the 5750 Gg of methane compare to the EPA GHG inventory? If they are the same, I'd cite the GHG inventory. If they are different, I'd still use the GHG inventory, but note the difference. Also, does this number include emissions from petroleum production as well as natural gas? Since you included associated gas in the leak rate calculation for the Eagle Ford on p. 11, line 27, the national leak rate should

also include emissions from petroleum production."

Response: The methane emissions estimate from the EPA NEI Oil and Gas Emissions tool is lower than that of the GHG inventory because the GHG inventory includes emissions from federal offshore waters (the Outer Continental Shelf) while the EPA NEI Oil and Gas Emissions tool does not. Since the Energy Information Administration includes natural gas from federal waters in their U.S. natural gas production estimate, the emissions estimate of 5,750 Gg from the EPA NEI Oil and Gas Emissions tool has been replaced by the GHG inventory estimate of 6,616 Gg of methane emissions in 2011. This number includes emissions from both oil and gas wells.

"p. 3, Line 16, This emission rate is somewhat misleading. Schneising et al. reported an energy content leak rate, which is not the same as a natural gas leak rate. The energy content leak rate takes into account the oil produced as well as the natural gas. See Howarth [Energy and Emission Control Technologies, 2015, p. 48] or Peischl et al. [JGR-Atmospheres, 2016, p. 2] for a discussion on this issue."

Response: We agree that the context of an emission rate must be explicit. We chose to express the emission rate of natural gas as a fraction of produced natural gas to be consistent with most studies in other shale areas, including dry shale basins such as the Marcellus, as well as bottom-up greenhouse gas inventories (e.g. EPA GHG). We have revised the manuscript to clarify the context of the emission rate. If we were to compare the methane emissions to produced energy following Scheising et al., our emission rate would be lower as the produced energy in the Eagle Ford is largely in the form of oil. However, note that the combined emissions of ethane, propane, and butanes exceed the mass of methane emissions in our study. This suggests that a comparison of methane emissions alone to produced energy content may not be appropriate because it excludes the energy emitted in the form of non-methane VOCs. We consider this to be beyond the scope of this manuscript.

"p. 4, Line 21, Why did you use the EDAS 40 km dataset for meteorology over others?

[Figure]

Perhaps include a sentence explaining your choice."

Response: This dataset was chosen for computational efficiency while running the HYSPLIT trajectory model over a set of several years. Despite its relatively coarse grid, it will capture the general synoptic-scale flow, so it is sufficient to use to identify days with southeasterly flow. A sentence was added to the text. Note that a more robust meteorological dataset (the NARR) was used to calculate emissions.

"p. 5, Line 1, Please show a time series of the background upwind mixing ratios and the enhanced mixing ratios at Floresville. This will give the reader a sense of how well the background sites represent the background air impacting the Floresville site."

Response: A timeline of the ethane mixing ratios at the upwind site in Corpus Christi and the downwind site in Floresville has been added to the supplemental information document. This figure also shows the seasonality of ethane which was questioned in another comment.

"p. 7, Line 7, Please provide some discussion of the PBL height and what effect the uncertainty of the modeled PBL height has on the analysis. Has the modeled PBL height been verified using LIDAR or aircraft measurements?"

Response: The planetary boundary layer height does serve as a source of uncertainty. The uncertainty was estimated using the spatial variability of the PBL height in a subset of grid cells in between the upwind and downwind sites. Supporting Table S3 shows the height of the PBL for each afternoon and the standard deviation of the aforementioned cells. This standard deviation was used to introduce uncertainty into the Monte Carlo simulation. On average, the PBL height was 1789 m with a standard deviation of 164 m. This is a relative standard deviation of 9.2%. That is small compared to the relative uncertainty of the ethane enhancement and the alkane composition in the raw natural gas and tank gas samples. A few sentences have been added to the manuscript to emphasize the uncertainty in the meteorological variables. A reference has also been added for a study in which the NARR PBL heights were shown to have no strong

bias compared to objectively determined PBL heights from sounding data, though the correlations were moderate and seasonally dependent.

"p. 9, Line 1, Did you not see a seasonal change in background ethane due to greater chemical loss during the summer?"

Response: The timeline of the ethane mixing ratios at the upwind and downwind sites shows the seasonality of background ethane mixing ratios. A brief discussion has been added to the text.

"p. 11, Line 33, A comparison with the EPA inventory estimate from petroleum production would be a fairer one, considering how much oil is produced in the Eagle Ford shale."

Response: Please see the response for the comment pertaining to p. 3, Line 16.

"Conclusions section, Please include a time frame for your emissions estimates. Are they for the entire study period? If so, please state this explicitly in the Conclusions."

Response: The emissions were estimated for a set of 68 days from August 2013 through August 2015. This was added to the text.

Grammar suggestions:

"p. 3, Lines 7-10, This is a long sentence. Consider splitting it up into two."

Response: Done

"p. 10, Line 1, I'm not sure "constraint" is a verb, unless it is an old-timey past tense."

Response: This was fixed in the text.

[revised manuscript text omitted]

**Introduction**

This document contains a description of the methodology used to bin HYSPLIT trajectories based on air mass origins in Text S1, with Table S2 and Figure S1 outlining the polygons used to test the paths of trajectories. Additionally, TCEQ sites where VOC data are collected are described in Table S1 and input variables for the Monte Carlo simulation are outlined in Table S3. Figures S2 through S7 support the results in the main text.

**Text S1.** In order to identify days with mostly southeasterly flow, 48 hour back trajectories were obtained from the HYSPLIT model. The start times for these trajectories were 06:00, 12:00, and 18:00 UTC each calendar day, and 00:00 UTC the following calendar day. The origins of the trajectories were binned by their passage through a series of polygons, as shown in Figure S1. The vertices of the polygons are provided in Table S2.

Back trajectories ending at the San Antonio – Old Highway 90 site were used to identify southeasterly flow when assessing long term alkane trends at San Antonio – Old Highway 90, Clute, and Corpus Christi – Hillcrest. All trajectories that passed through Polygon 1 were assumed to have continental origins and were removed. Polygon 2 was selected to represent the central Texas Coast region, roughly extending from the coastal waters southeast of Corpus Christi to the waters south of Clute. Trajectories that did not pass through this polygon were also removed, leaving generally southeasterly trajectories of maritime origin remaining. Days with 3 out of 4 southeasterly trajectories were used to compare long term alkane trends at these sites.

For the alkane emission calculation using data from Corpus Christi – Oak Park and Floresville, back trajectories ending at Floresville were removed if they passed through Polygon 1. Again, this was to remove air masses which were influenced by continental emissions prior to moving ashore. Trajectories were also removed if they did not pass through Polygon 3, which encompasses Corpus Christi and the surrounding region of the Texas Coast. Again, days with 3 out of 4 trajectories were used to quantify the afternoon alkane enhancement between Corpus Christi – Oak Park and Floresville.

**Table S1.** Vertices of polygons used to bin HYSPLIT trajectories as described in Text S1.

| Polygon | Vertex 1 | Vertex 2 | Vertex 3 | Vertex 4 | Vertex 5 | Vertex 6 |
|---|---|---|---|---|---|---|
| 1 | 29.16° N, 96.12° W | 29.16° N, 83.00° W | 31.00° N, 83.00° W | 31.00° N, 103.00° W | 29.50° N, 103.00° W | 29.50° N, 96.12° W |
| 2 | 27.04° N, 96.75° W | 27.43° N, 97.07° W | 28.67° N, 95.50° W | 28.28° N, 95.19° W | - | - |
| 3 | 27.27° N, 97.48° W | 27.81° N, 98.00° W | 28.16° N, 97.64° W | 27.62° N, 97.12° W | - | - |

**Table S2.** Separate Excel file showing average alkane mixing ratios (ppb) during afternoon hours at Floresville and Corpus Christi – Hillcrest during 68 days with appropriate meteorological conditions. Also shown are input meteorology from the North American Regional Reanalysis (NARR) that were used as input for the mass balance approach and Monte Carlo simulation. Lastly, alkane emission rates and methane emissions relative to production are provided for each day.

[Figure]

**Figure S1.** Polygons used to identify trajectories as southeasterly with maritime origins. See Text S1 for a description of the methods and Table S2 for corners of the vertices.

[Figure]

**Figure S2.** Scatterplot of afternoon ethane mixing ratios at the upwind site (Oak Park), the downwind site (Floresville) over 68 days with southeasterly flow. Note that each mixing ratio has a relative uncertainty of ±5.8% and the uncertainty of the enhancement is equal to the sum of the uncertainties of the upwind and downwind sites. The background colors show the warm and cool seasons. Ethane mixing ratios are generally lower at both the upwind and downwind sites during the summer and fall and higher in the winter and spring.

[Figure]

**Figure S3.** Scatterplot of afternoon isopentane and *n*-pentane enhancements observed between Oak Park and Floresville over 68 days with southeasterly flow. Also shown are isopentane and *n*-pentane mixing ratios in San Antonio for 24 hour canister samples from 2007 to 2015. The slopes of both linear regressions are highly statistically significant ($p < 0.001$). The isopentane-to-n-pentane ratio in San Antonio is 2.25, which is indicative of urban emissions. Meanwhile, the slope of the ratio for the enhancements is 1.17, indicating emissions from petroleum production.

[Figure]

**Figure S4.** Probability distribution functions for the fraction of observed alkane ratios that can be explained by emissions of raw natural gas ($f_{RNG}$). Note that the x-axis extends to numbers that are physically unreasonable. This is due to the assumption that alkane ratios can be explained by emissions from raw natural gas and vented tank gas alone (the compositions of which are highly uncertain), and other sources and sinks are neglected. Nonetheless, there is general agreement between the fractions derived from the $C_3/C_2$ ratio, the $nC_4/C_2$ ratio, and the $iC_4/C_2$ ratio, suggesting that the above assumption creates results that are reproducible using these three alkane ratios.

[Figure]

**Figure S5.** Distributions of the 25th, 50th, and 75th percentile of relative emissions over all 68 days. Methane emissions were converted to a volume of natural gas using methane to natural gas ratios (Table 1 in main text) and compared to natural gas production. The top panel shows total methane emissions while the bottom panel shows only emissions from raw natural gas (RNG) sources and excludes emissions from liquid storage tanks. Some outliers of the lower

bound were less than zero due to large uncertainties in the methane enhancement which occasionally produced a negative methane flux within the Monte Carlo simulations. The  total emission rate is often close to that of the EPA/EIA emission estimate for nationwide natural gas and associated gas production in 2011 . The emission rate for raw natural gas emissions alone was generally lower than the EPA/EIA estimate. Note that the emission rate estimate has a slightly skewed distribution, with the upper bound possibly influenced by plumes from large emitters.

[Figure]

**Figure S6.** Timeline of the median emission rate for each day with the interquartile range represented by whiskers. The emission rate showed neither apparent seasonality nor trend over time, which is to be expected of a continuous emission source that does not depend on meteorological variables.

[Figure]

**Figure S7.** The median emission rate for each of the 68 days plotted against wind speed (component parallel to the transect between Corpus Christi and Floresville), PBL depth, temperature, and ethane enhancement. While there were statistically significant correlations with wind speed, temperature, and ethane enhancement, these correlations were week. While wind speed shared the strongest correlation with the emission rate ($R^2 = 0.13$), this suggests that the emission rate was not strongly driven by meteorological variability.

[Figure]

**Figure S8.** Example of a tornado plot from a Monte Carlo simulation for 2 August 2013. The total emission rate depended largely on the composition of raw natural gas (RNG), with lower ethane content in natural gas resulting in a higher emission rate. The RNG-only emission rate shows a negative correlation with the ethane content in both RNG and vented gas from liquid storage tanks (TG), as these numbers were used to partition emissions between RNG and TG sources. Both the total and RNG-only emission rates showed positive correlations with wind speed and planetary boundary layer (PBL) depth, but not temperature. The rates were also positively correlated with the ethane enhancement.

---

## Author Response (AR2)

To the editor,

We identified a discrepancy between the calculation code and the text in our manuscript. While that discrepancy has been fixed, we would like to inform you that the resulting change altered one of our conclusions.

In order to address the most recent Reviewer's comment regarding the uncertainty in the planetary boundary layer (PBL) height, we edited our code regarding the PBL depth in the Monte Carlo simulation. During this editing, we noted that the relative emission rates (expressed as a percentage of produced natural gas) were calculated using ethane emissions while the manuscript indicated that the emission rates were calculated using estimated methane emissions. We fixed the related code to estimate the relative emission rates using methane emissions to match the manuscript.

This change did not impact the raw-natural-gas-only relative emission rate of 0.7% (interquartile range of 0.5 to 1.3%) of produced natural gas. However, the corrected total relative emission rate is 1.0% (0.7 to 1.6%). We find that the median emission rate is lower than the EPA's current emissions estimate, which is approximately 1.6% of produced gas. This changes the conclusion in the previous draft of our manuscript, which stated that the median total emission estimate of 2.1% exceeded the EPA's emission estimate. Our results and conclusions regarding emissions for non-methane VOCs remain unchanged.

We apologize for this issue and understand if additional reviews are necessary.

Thank you,

Geoff Roest and Gunnar Schade

**Minor revisions on "Quantifying alkane emissions in the Eagle Ford Shale using boundary layer enhancement" by G. Roest and G. Schade**

Anonymous Referee #2

The authors have addressed most of the issues raised by the original reviewers. However, one main issue remains that must be addressed before I would recommend publication. The issue I have is with the authors' treatment of the PBL height and its related uncertainties. In their response to reviewers, page C5, the authors refer to an average PBL height of 1789 ± 164 m, or an uncertainty of 9.2%. First, in the supplementary materials provided, there is no Table S3, so it is difficult to ascertain the authors' treatment of the PBL height uncertainty. It is also not explicitly stated in the text, which it probably should be. Second, for the Barnett region, Karion et al. (2015) estimated the PBL height uncertainty between 12 and 31%, with an average of 24%, and that was from actual measurements of the PBL height. I find it difficult to believe that the NARR model would be more than twice as certain as actual measurements, especially since both study regions are relatively flat. I think the modeled PBL height should have at least a 25% uncertainty, if not more. While increasing the PBL height in the Monte Carlo analysis likely won't change the conclusions of the manuscript (although the IQR of the emission rate will likely increase), I think it would make it more defensible scientifically.

Response: We used the standard deviation of the NARR PBL depth over a 1° latitude by 1° longitude box situated upwind of Floresville as a proxy for the uncertainty in the PBL depth. This is discussed in Section 2.5. We are unaware of an uncertainty that is assigned to the PBL depth by the NARR team. However, we agree that the uncertainty in the NARR's PBL depth should be increased in the analysis, especially because it has been shown to contribute largely to the uncertainty in other mass balance studies.

We are aware of two studies that have compared the NARR PBL depths to objectively-analyzed sounding data. A study at a field site in the Southern Plains (Schmid and Niyogi, 2012), which has already been reference in the manuscript, shows that there is a modest correlation (r = 0.39 to 0.58) and no strong bias in the NARR PBL depth when compared to objectively analyzed PBL depths using radiosonde data. A similar study in Virginia (Lee and De Wekker, 2016) shows that the standard deviation between the PBL depths estimated using a Bulk Richardson method with the NARR data and the objectively analyzed radiosonde PBL depths was 453.1 meters. While this study also showed a bias, we have not added a bias to the NARR PBL depth in the Monte Carlo estimate as this study used a different method to estimate the PBL height.

We have increased the uncertainty of the PBL depth in the Monte Carlo simulations, which is now represented by a standard deviation of 500 meters. On average, this value represents more than 25% of the PBL depth in the 1° latitude by 1° longitude box. This change did slightly increase the IQR of alkane emission rates within each Monte Carlo simulation, and the emission rates now show a stronger correlation with PBL depth.

References:

Lee, T. R., and S. F. J. De Wekker (2016), Estimating Daytime Planetary Boundary Layer Heights over a Valley from Rawinsonde Observations at a Nearby Airport: An Application to the Page Valley in Virginia, United States, *Journal of Applied Meteorology and Climatology*, *55*(3), 791–809, doi:10.1175/JAMC-D-15-0300.1.

Schmid, P., and D. Niyogi (2012), A Method for Estimating Planetary Boundary Layer Heights and Its Application over the ARM Southern Great Plains Site, *Journal of Atmospheric and Oceanic Technology*, *29*(3), 316–322, doi:10.1175/JTECH-D-11-00118.1.

Other comments:

page 1, line 11: uncapitalize n-Butane

Response: This comment has been accepted and addressed in the text.

page 2, lines 16 to 21: the volumes reported should be in SI units

Response: The standard units for reporting natural gas production in the U.S. are cubic feet. Nonetheless, this comment has been accepted and addressed in the text.

page 5, line 34: can the authors quantify the "major source of uncertainty" that has been introduced? How do the authors know it is "major"? If the authors ignore it, then it really introduces no additional uncertainty other than what are stated in the conclusions. If the authors accounted for this uncertainty, they should state what it is. Otherwise, I would change "a major source of uncertainty" to "an unquantified uncertainty".

Response: The authors agree that, due to the lack of data on gas stream composition from the Eagle Ford Shale, the uncertainty introduced by using data from the Barnett Shale is unquantified. This comment has been accepted and addressed in the text.

page 7, line 17: remove "the" to read "as outlined in Sect. 2.4."

Response: This comment has been accepted and addressed in the text.

Figure 2: why wouldn't ethane mixing ratios from the south be expected to be as great as when the winds are from the southeast or west?

Response: Winds at Floresville that are out of the E to ESE or SW to WSW suggest that trajectories may flow across the Eagle Ford Shale at an angle that allows for an accumulation of emissions over a longer period of time as compared to when winds are blowing out of the S to SE. In the latter case trajectories may only briefly be exposed to emissions in the Eagle Ford as air passes nearly perpendicular to the axis of the shale,

leaving less time for pollutants to accumulate in the air mass. This explanation has been added to the figure caption.

Figure 4: does this map include the county that Floresville is in, and is it colored on the grey scale? If not, I'd add it; if so, I'd state that in the caption and in the main body where the discussion of how far Floresville is from the nearest well.

Response: Wilson County, where Floresville is located, is shaded in this map, but its natural gas production is so low compared to other counties included in this map (<0.1% of Karnes County on a pixel basis) that it appears white even though the shading is non-linear. As stated in the figure caption: "The grey shading underlying the plume map identifies counties with the sum of natural gas and associated gas production, darker shading indicating higher production rates. The darkest shading indicates Karnes County, not labeled for clarity." However, for increased clarity, we have included the county boundaries and the Wilson County label in an updated Figure.

Figure S7: change to "correlations were weak"

Response: This comment has been accepted and addressed in the caption.

Additional change:

We identified a discrepancy between the calculation code and the text in our manuscript. In our Monte Carlo simulation code, the relative emission rates (expressed as a percentage of produced natural gas) were calculated using ethane emissions while the manuscript indicated that the emission rates were calculated using estimated methane emissions. We fixed the related code to estimate the relative emission rates using methane emissions to match the manuscript. This change did not impact the raw-natural-gas-only relative emission rate of 0.7% (interquartile range of 0.5 to 1.3%) of produced natural gas. However, the corrected total relative emission rate is 1.0% (0.7 to 1.6%).

[revised manuscript text omitted]

**Introduction**

This document contains a description of the methodology used to bin HYSPLIT trajectories based on air mass origins in Text S1, with Table S2 and Figure S1 outlining the polygons used to test the paths of trajectories.  Additionally, TCEQ sites where VOC data are collected are described in Table S1 and input variables for the Monte Carlo simulation are outlined in Table S3.  Figures S2 through S7 support the results in the main text.

**Text S1.** In order to identify days with mostly southeasterly flow, 48 hour back trajectories were obtained from the HYSPLIT model.  The start times for these trajectories were 06:00, 12:00, and 18:00 UTC each calendar day, and 00:00 UTC the following calendar day.  The origins of the trajectories were binned by their passage through a series of polygons, as shown in Figure S1.  The vertices of the polygons are provided in Table S2.

Back trajectories ending at the San Antonio – Old Highway 90 site were used to identify southeasterly flow when assessing long term alkane trends at San Antonio – Old Highway 90, Clute, and Corpus Christi – Hillcrest.  All trajectories that passed through Polygon 1 were assumed to have continental origins and were removed.  Polygon 2 was selected to represent the central Texas Coast region, roughly extending from the coastal waters southeast of Corpus Christi to the waters south of Clute.  Trajectories that did not pass through this polygon were also removed, leaving generally southeasterly trajectories of maritime origin remaining.  Days with 3 out of 4 southeasterly trajectories were used to compare long term alkane trends at these sites.

For the alkane emission calculation using data from Corpus Christi – Oak Park and Floresville, back trajectories ending at Floresville were removed if they passed through Polygon 1.  Again, this was to remove air masses which were influenced by continental emissions prior to moving ashore.  Trajectories were also removed if they did not pass through Polygon 3, which encompasses Corpus Christi and the surrounding region of the Texas Coast.  Again, days with 3 out of 4 trajectories were used to quantify the afternoon alkane enhancement between Corpus Christi – Oak Park and Floresville.

**Table S1.** Vertices of polygons used to bin HYSPLIT trajectories as described in Text S1.

| Polygon | Vertex 1 | Vertex 2 | Vertex 3 | Vertex 4 | Vertex 5 | Vertex 6 |
|---|---|---|---|---|---|---|
| 1 | 29.16° N, 96.12° W | 29.16° N, 83.00° W | 31.00° N, 83.00° W | 31.00° N, 103.00° W | 29.50° N, 103.00° W | 29.50° N, 96.12° W |
| 2 | 27.04° N, 96.75° W | 27.43° N, 97.07° W | 28.67° N, 95.50° W | 28.28° N, 95.19° W | - | - |
| 3 | 27.27° N, 97.48° W | 27.81° N, 98.00° W | 28.16° N, 97.64° W | 27.62° N, 97.12° W | - | - |

**Table S2.** Separate Excel file showing average alkane mixing ratios (ppb) during afternoon hours at Floresville and Corpus Christi – Hillcrest during 68 days with appropriate meteorological conditions. Also shown are input meteorology from the North American Regional Reanalysis (NARR) and production data from the Railroad Commission (RRC) of Texas that were used as input for the mass balance approach and Monte Carlo simulation. Lastly, alkane emission rates and methane emissions relative to production are provided for each day.

[Figure]

**Figure S1.** Polygons used to identify trajectories as southeasterly with maritime origins. See Text S1 for a description of the methods and Table S2 for corners of the vertices.

[Figure]

**Figure S2.** Scatterplot of afternoon ethane mixing ratios at the upwind site (Oak Park), the downwind site (Floresville) over 68 days with southeasterly flow. Note that each mixing ratio has a relative uncertainty of ±5.8% and the uncertainty of the enhancement is equal to the sum of the uncertainties of the upwind and downwind sites. The background colors show the warm and cool seasons. Ethane mixing ratios are generally lower at both the upwind and downwind sites during the summer and fall and higher in the winter and spring.

[Figure]

**Figure S3.** Scatterplot of afternoon isopentane and *n*-pentane enhancements observed between Oak Park and Floresville over 68 days with southeasterly flow. Also shown are isopentane and *n*-pentane mixing ratios in San Antonio for 24 hour canister samples from 2007 to 2015. The slopes of both linear regressions are highly statistically significant ($p < 0.001$). The isopentane-to-n-pentane ratio in San Antonio is 2.25, which is indicative of urban emissions. Meanwhile, the slope of the ratio for the enhancements is 1.17, indicating emissions from petroleum production.

[Figure]

[Figure]

**Figure S4.** Probability distribution functions for the fraction of observed alkane ratios that can be explained by emissions of raw natural gas ($f_{RNG}$). Note that the x-axis extends to numbers that are physically unreasonable. This is due to the assumption that alkane ratios can be explained by emissions from raw natural gas and vented tank gas alone (the compositions of which are highly uncertain), and other sources and sinks are neglected. Nonetheless, there is general agreement between the fractions derived from the $C_3/C_2$ ratio, the $nC_4/C_2$ ratio, and the $iC_4/C_2$ ratio, suggesting that the above assumption creates results that are reproducible using these three alkane ratios.

[Figure]

[Figure]

**Figure S5.** Distributions of the 25[th], 50[th], and 75[th] percentile of relative emissions over all 68 days. Methane emissions were converted to a volume of natural gas using methane to natural gas ratios (Table 2 in main text) and compared to natural gas production. The top panel shows total relative emission rate while the bottom panel shows only emissions from raw natural gas (RNG) sources and excludes emissions from liquid storage

tanks. Some outliers of the lower bound were less than zero due to large uncertainties in the methane enhancement which occasionally produced a negative methane flux within the Monte Carlo simulations. The  upper bound of the total emission rate is often close to that of the EPA/EIA emission estimate for nationwide natural gas and associated gas production in 2011 . The emission rate for raw natural gas emissions alone was generally lower than the EPA/EIA estimate. Note that the emission rate estimate has a slightly skewed distribution, with the upper bound possibly influenced by plumes from large emitters.

[Figure]

[Figure]

**Figure S6.** Timeline of the median emission rate for each day with the interquartile range represented by whiskers. The emission rate showed neither apparent seasonality nor trend over time, which is to be expected of a continuous emission source that does not depend on meteorological variables.

[Figure]

[Figure]

**Figure S7.** The median emission rate for each of the 68 days plotted against wind speed (component parallel to the transect between Corpus Christi and Floresville), PBL depth, temperature, and ethane enhancement. While there were statistically significant correlations with wind speed, temperature, and ethane enhancement, these correlations were weak. While the PBL depth showed the strongest correlation with the emission rate ($R^2$ = 0.18, p < 0.001), the emission rate was not strongly driven by meteorological variability.

[Figure]

**Figure S8.** Example of a tornado plot from a Monte Carlo simulation for 2 August 2013. The total emission rate depended largely on the composition of raw natural gas (RNG), with lower ethane content in natural gas (which was used, in part, to estimate methane emissions) resulting in a higher emission rate. The RNG-only emission rate shows a negative correlation

with the ethane content in both RNG and vented gas from liquid storage tanks (TG), as these numbers were used to partition emissions between RNG and TG sources. Both the total and RNG-only emission rates showed positive correlations with wind speed and planetary boundary layer (PBL) depth, but not temperature. The rates were also positively correlated with the

5   ethane enhancement.

---

## Author Response (AR3)

"Quantifying alkane emissions in the Eagle Ford Shale using boundary layer enhancement" by G. Roest and G. Schade

The following technical changes requested by the co-editor were addressed. No other changes have been made.

Co-editor comments:

Please make the following technical changes:

page 9, line 8: put 'e.g.,' before 'Karion et al.'

    Response: This comment has been accepted and addressed in the text.

page 12, line 25: this text could be improved for clarity. Perhaps " ... – a variable with large relative uncertainty that was used to estimate methane emissions –"

    Response: This comment has been accepted and addressed in the text.

page 13, line 6: you don't need to refer to Table 2 here in the Conclusions. This should already have been addressed in the text. Simply state the ratio.

    Response: This comment has been accepted and addressed in the text.

[revised manuscript text omitted]